# S³: Increasing GPU Utilization during Generative Inference for Higher Throughput

**Yunho Jin**
Harvard University

**Chun-Feng Wu**
National Yang Ming
Chiao Tung University

**David Brooks**
Harvard University

**Gu-Yeon Wei**
Harvard University

## Abstract

Generating texts with a large language model (LLM) consumes massive amounts of memory. Apart from the already-large model parameters, the key/value (KV) cache that holds information about previous tokens in a sequence can grow to be even larger than the model itself. This problem is exacerbated in one of the current LLM serving frameworks which reserves the maximum sequence length of memory for the KV cache to guarantee generating a complete sequence as they do not know the output sequence length. This restricts us to use a smaller batch size leading to lower GPU utilization and above all, lower throughput. We argue that designing a system with a priori knowledge of the output sequence can mitigate this problem. To this end, we propose S³, which predicts the output sequence length, schedules generation queries based on the prediction to increase device resource utilization and throughput, and handle mispredictions. Our proposed method achieves $6.49\times$ throughput over those systems that assume the worst case for the output sequence length.

## 1 Introduction

Text generation has become increasingly popular in various services, leading to a surge in the use of large language models (LLMs) known for their high accuracy. LLMs have distinct features that differentiate them from non-Transformer-based models, including a requirement for massive amounts of compute and memory resources. Specifically, we observe that Transformer-based LLMs are often limited by memory capacity and bandwidth, resulting in significant underutilization of compute resources. When serving GPT-J on an NVIDIA A100 GPU, the utilization of GPU compute resources can be as low as 0.4%. This highlights the memory-bound nature of the LLMs and the need for efficient memory utilization to increase the utilization of GPU compute resources.

A common approach to boost GPU utilization and enhance throughput is to increase batch size. This is due to the fact that inputs within a batch share the same model weights, thus the GPU only needs to load the model weight from its high bandwidth memory (HBM) to the on-chip SRAM once and reuse it for all inputs within the batch. The GPU uses more of its compute resources when processing the same model weights. Increasing batch size is a simple optimization technique and is, therefore, commonly used in serving convolutional and fully-connected neural network models [1, 2].

However, the self-attention layer in Transformer-based text generation LLMs presents a challenge to this simple optimization due to its autoregressive nature. Specifically, when generating a new token in a sequence, the model needs to attend to all previous tokens in the sequence, requiring the model to retain all information from previous tokens and store them in HBM. We call this region in the HBM holding the information key/value cache (KV cache). The size of the KV cache grows with larger batch sizes and longer sequences, which limits the maximum batch size, thereby lowering GPU utilization and ultimately reducing throughput. To support the growing KV cache size, Huggingface's Transformers library [3] constantly allocates new memory at each token generation

37th Conference on Neural Information Processing Systems (NeurIPS 2023).

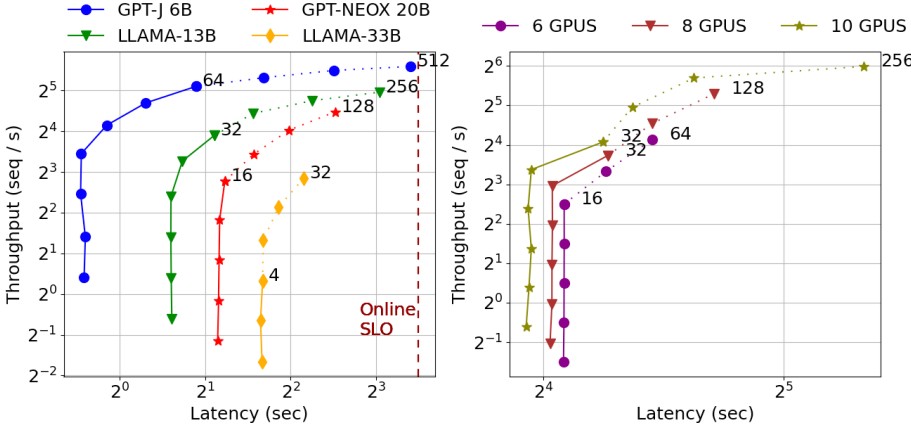

Figure 1: Latency versus throughput trade-off among different models (left, online scenario with single GPU running models within the size confines of the GPU) and the number of GPUs (right, offline scenario distributing GPT-3 175B to 6, 8, and 10 GPUs) when generating 60 tokens, inspired by FlexGen [6]. The markers in the lines represent batch sizes, from 1 to the maximum batch size that can be loaded on an A100 GPU, incrementing by the power of two. Allocating the exact amount of memory for each sequence expands the curve to higher throughput at the cost of higher latency. The solid lines show the trade-off in vanilla systems and dotted lines show how much $S^3$ can expand the trade-off. The numbers represent the maximum batch sizes for $S^3$ and vanilla systems. The vertical line in the left figure denotes the latency SLO for reading a 60-token-long sequence.

and incurs latency overhead associated with memory allocation. This improves usability since users do not have to know the output sequence length but suffers long inference latency. Alternatively, NVIDIA's FasterTransformer library [4] pre-allocates memory for the maximum sequence length, which ends up wasting memory by allocating more than is necessary. This approach limits batch size and prioritizes latency over throughput. The trend towards long maximum sequence lengths in recent models (e.g., 8K or 32K) [5] amplifies the KV cache overhead, demanding more efficient memory utilization in serving Transformer-based text generation models.

In light of these challenges, we propose $S^3$, scheduling sequences with speculation, a framework that maximizes throughput via predicting the output sequence length and reducing memory waste. The frequent memory allocations in Huggingface's Transformers and the limited batch size in Faster-Transformer stem from the fact that we lack prior knowledge of the output sequence length. $S^3$ addresses these issues by predicting the expected output sequence length and allocating the corresponding amount of memory for each query. It also schedules sequences to increase the GPU utilization. Finally, $S^3$ runs a supervisor in the background that detects mispredictions and adjusts the size of allocated memory to be more accurate. By integrating these components together, $S^3$ optimizes memory usage and scheduling to maximize throughput during deployment of Transformer-based LLMs for text generation on GPUs.

There are two types of LLM deployment scenarios: online and offline. Online scenarios such as chatbots [7, 8] require service providers to generate a sequence within a tight latency service level objective (SLO) constraint. Offline scenarios include applications such as scoring [9] or data wrangling [10], and have loose latency SLOs, emphasizing throughput over end-to-end latency. In contrast, FaterTransformers [4] and xFormers [11] prioritize reducing latency. We argue that ensuring the latency remains below the SLO constraint renders it unnecessary to prioritize further latency reduction. To this end, we design $S^3$ to achieve higher throughput under those latency SLOs. Figure 1 highlights how much $S^3$ can improve throughput when trading off latency. For online scenarios, we assume a latency SLO set to the average reading speed of English readers, 4 words per second [12] and 0.1875 second per token [13]. The models on the left figure that are smaller than 100 billion parameters satisfy this SLO for all possible batch sizes. This SLO offers service providers to improve throughput over all models. In fact, for LLAMA-33B [14], there are opportunities to get throughput benefits with no latency penalties. The right figure, sweeping over different number of GPUs, shows opportunities to maximize throughput in offline scenarios that have loose latency SLO.

We evaluate $S^3$, assessing both its throughput and cost-efficiency. Our analysis includes both offline and online scenarios. In online scenarios under the average reading speed latency SLO constraint,

we find that S$^3$ can generate up to 6.49× more sequences while adhering to the same SLO constraint. In offline scenarios, we observe that S$^3$ achieves a speedup up to 6.49× for different models. S$^3$ does not affect the models' perplexity as it does not change the models' architectures. Furthermore, we evaluate the cost-efficiency of S$^3$ and find that using 6 GPUs, S$^3$ provides almost identical throughput compared to a vanilla system with 10 GPUs.

To summarize, we make the following contributions:

- We increase the achievable batch size under longer latency SLO and allow service providers to serve with higher throughput in both online and offline scenarios by using larger batch sizes.
- We fine-tune a Distillbert model to predict output sequence lengths given an input prompt.
- We provide a mechanism to recover from mispredictions. S$^3$ preempts sequences that exceed their allocated memory and retrain the predictor to learn from its mistakes.

## 2 Background and Motivation

### 2.1 Generative AI Models

A Transformer-based generative model is autoregressive. It predicts the most probable token based on past tokens. Since the model generates one token at a time, it has to iterate over itself $n$ times to generate a sequence that is $n$-tokens long. One iteration involves an input token traversing through the model which is a stack of transformer layers containing one attention, two layer norm, and two feed-forward layers. Especially, the self-attention layer uses information on the past tokens to generate the next token.

For example, the model at the $i^{th}$ iteration attends the current token ($t_i$) with every token it already generated ($t_0, ...t_{i-1}$) in the self-attention layer. We can express the self-attention layer as:

$$h_{out} = softmax(\frac{q_i \cdot K^T}{\sqrt{d_h}}) \cdot V$$

where $d_h$ is hidden dimension of the model, $h_{out}, q_i \in \mathbb{R}^{d_h}$ are output hidden vector, current query vector, respectively, and $K, V \in \mathbb{R}^{i \times d_h}$ are key and value matrices. The $j_{th}$ rows in the $K$ and $V$ matrices represent key and value vectors of $t_j$, respectively. The two dot products attend $t_i$ to all the key and value vectors in the current sequence.

The model stores $K$ and $V$ matrices as the key/value (KV) cache to avoid having to generate key and value vectors at each iteration. Otherwise, it has to store hidden states of every previous token and multiply it with a weight matrix $W_{QKV} \in \mathbb{R}^{d_h \times 2d_h}$ at every transformer layer. This would require $2(i-1)d_h$ FLOPs per layer almost identical to $2id_h$ FLOPs for the self-attention layer at $t_i$. The size of the KV cache is $4ld_h$ bytes per token when using half-precision numbers. The cache uses 2 bytes for every number for both the key and value cache where $l$ is the number of transformer layers in a model. For example, GPT-NEOX [15], a 20 billion parameter model, has 44 layers and 6144 hidden dimensions and thus uses 1MB per KV cache per token.

### 2.2 KV Cache Management on GPUs

The KV cache is relatively small (e.g., several MBs) and can be easily stored in the GPU HBM (high-bandwidth memory) when the sequence is short as the cache stores information about the previous tokens in the sequence. It grows as the model generates more tokens. Using this dynamic nature, Huggingface's Transformers [3] (HF-Transformers) constantly allocates more memory to the KV cache and stalls the computation until it is complete. This approach allows the library to allocate the exact amount of memory for each cache at the cost of frequent memory accesses.

To mitigate this, NVIDIA's FasterTransformer library reserves the maximum sequence length of memory for every sequence [4, 16]. It removes redundant memory accesses by simply filling in the reserved memory in an append-only fashion. However, this approach comes with its own drawback as it reserves more than strictly-necessary memory for sequences. For GPT-NEOX with a maximum sequence length of 2048 tokens, FasterTransformer reserves 2048 tokens even for sequences that

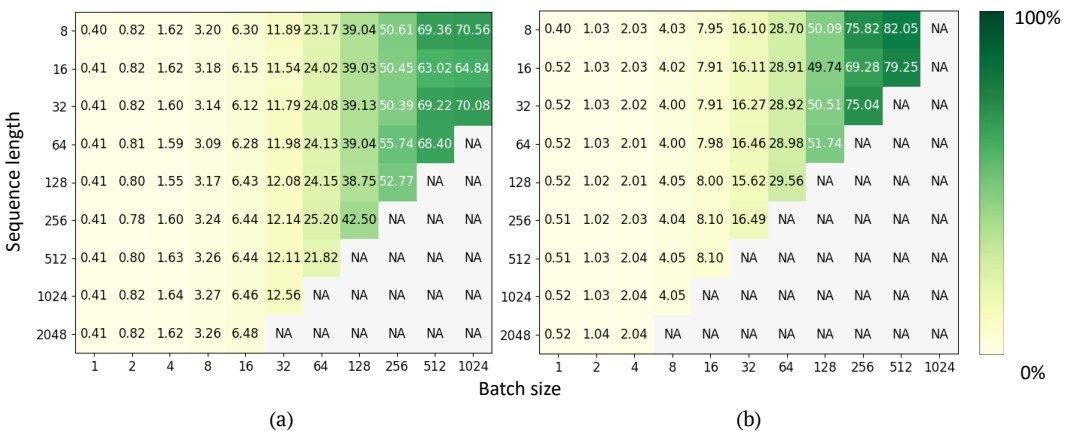

Figure 2: (a) GPT-J's and (b)GPT-NEOX's NVIDIA A100 compute utilization.

end up being 50 tokens long. The maximum batch size that FasterTransformer can use for this model with a 80GB A100 GPU is less than 20 with the 40GB model size and 2.2GB per sequence. The small batch size underutilizes the massive compute resources in the GPU. The rationale for reserving the maximum sequence length amount of memory even with the underutilization problem is to ensure that it generates full sequences and enhances user experience.

## 2.3 Observation

**Language models are memory bound**  We demonstrate the extent of GPU resource underutilization when we run GPT-J with 6B parameters on an A100 GPU. The relatively smaller model shows a wider spectrum of different batch sizes and sequence lengths since we can fit larger batches with longer sequences in the GPU. Fig. 2 (a) shows the GPU utilization swept over different batch sizes and sequence lengths. As the figure denotes, increasing the batch size achieves a higher utilization but eventually faces a memory cliff, where an out-of-memory (OOM) error kills the process. Take the batches with 1024 sequence length for example. 32 sequences are the maximum batch size and thus 12.56% is the maximum utilization that we can achieve with this sequence length. Fig. 2 (b) shows similar underutilization in the larger GPT-NEOX model due to the memory cliff problem. This model faces the memory cliff with smaller batch sizes and shorter sequences since the model consumes more of the GPU memory. Please note that HF-Transformers still needs to know the output sequence length before batching inputs to avoid the memory cliff.

As the figures illustrate, increasing the batch size can enhance throughput in neural networks. This approach does not require intricate optimizations and enables the GPU to load the model weight from its HBM to the on-chip SRAM only once and share it among a larger number of inputs. By doing so, the GPU can activate its idle compute resources and concurrently handle multiple inputs. Nevertheless, the memory cliff poses a challenge, limiting the utilization of additional resources. However, as we elaborate, this issue can be resolved.

**Reasons behind the inefficiencies**  Both the frequent memory allocations in HF-Transformers and the limited batch size in FasterTransformer come from the fact that we are not aware of the generated sequence length. If we know the precise length of the generated sequence, we can allocate exact memory to each sequence and resolve the repetitive memory reservation and the unnecessary memory allocation problems in HF-Transformers and FasterTransformer, respectively.

## 3 $S^3$ Design

$S^3$ is a system-algorithm co-design framework that maximizes GPU utilization with sequence length prediction to achieve higher throughput. $S^3$ has three components as shown in Fig. 3: 1) predictor, 2) scheduler, and 3) supervisor. A text generation query arrives in a request pool in the host DRAM. The predictor then predicts its output sequence length which the scheduler uses to batch requests. The scheduler dispatches the batch to the GPU and the text generator model generates texts in the

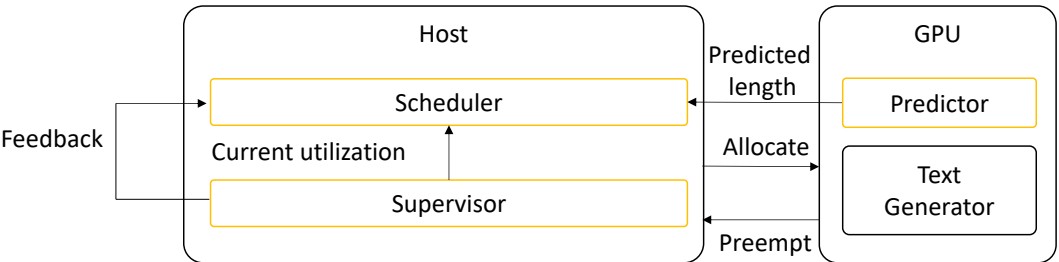

Figure 3: Overview of S$^3$. The boxes in yellow denote new components proposed by S$^3$.

batch. The supervisor oversees the GPU utilization and handles mispredictions. We describe each component in detail and how they interact with each other in this section.

**Output sequence length predictor**    We use a predictor to predict the output sequence length and resolve the frequent and redundant memory allocation problems. Specifically, we fine-tune a Distilbert [17] model that was trained for sequence classification to classify which length bucket the output sequence length falls into. We bucketize the sequence lengthes since it is known that machine learning models are not as capable of the "last mile" search compared to narrowing the candidates down to the last mile. Each bucket is allocated the range of $\frac{max\ sequence\ length}{number\ of\ buckets}$ and we use 10 buckets.

To this end, we fine-tune the model on the Alpaca dataset [18], one of the representative question-and-answering datasets, and use the questions as inputs and the lengthes of the answers as labels. We observe that this predictor predicts the correct bucket with 98.61% accuracy. The average distance between the wrong predictions and the correct bucket is 1.03 meaning that the error converges to 0 when we double the bucket size. We also evaluate the predictor on a model fine-tuned with Google Natural-Question dataset [19] and observe an accuracy of 77.13%. It makes smaller mistakes more often than larger ones. For completeness, we fine-tune a model on the Pile dataset [20], a non-question-and-answering dataset, and see 65.6% accuracy. The predictor shows surprisingly high accuracy compared to randomly guessing the bins as the latter is correct only 10% of the time.

We choose Distilbert, a 66 million parameter model for its small size and fast inference. The model size is negligible since it is smaller than even a single transformer layer in the billion-scale models (e.g., 214 million for 6 billion GPT-J [21] model). The latency is also negligible since the predictor model runs only once when a request arrives at the server while the text generation model runs $n$ times to generate an $n$-token long output sequence. We measure that Distilbert takes 3.7ms to run compared to several seconds for the text generation models on an NVIDIA A100 GPU.

**Length-aware sequence scheduler**    The scheduler batches and schedules sequences based on the predicted results to maximize the GPU utilization without exceeding the GPU HBM capacity. We can formulate this problem as a variant of the bin packing problem with a single bin. The capacity bin is the HBM size and the item weight is the size of the KV cache for each sequence.

We use the decreasing first fit algorithm [22] as the solution to the bin packing problem for its simplicity. The scheduler queues the lengthiest sequences first, reserving room for shorter sequences within the GPU's available HBM. It orders sequences in the request pool by length in decreasing order and iterates through the pool to check if the KV cache of the current sequence does not exceed the available HBM. If so, it includes the sequence in the current batch and reduces the available HBM by the size of the KV cache. The scheduler continues this process until either there is no available HBM or it has iterated through the entire request pool. This approach has been consciously adopted due to its minimal associated overhead, while still maintaining an approximately-optimal resolution for the problem at hand.

The resulting batch is irregularly shaped where some sequences are longer than others. Unfortunately, current frameworks either do not support irregularly shaped batch [3, 4] or support it with limited performance [23]. Those that do not support this functionality pad the short sequences with padding tokens to match the length of sequences in the same batch. The padding tokens waste both computation and memory since they do not hold any useful information. ORCA [16] introduces an interesting solution to this problem termed selective batching. The authors of the work identify

Table 1: Model architecture used in the evaluations

| Model | Num Params | Num layers | Model dim | Num heads |
|---|---|---|---|---|
| GPT-J [21] | 6B | 28 | 5120 | 16 |
| LLAMA 13B [14] | 13B | 40 | 4096 | 40 |
| GPT-NEOX [15] | 20B | 44 | 6144 | 64 |
| LLAMA 33B [14] | 30B | 60 | 6656 | 52 |
| GPT3 175B [26] | 175B | 96 | 12288 | 96 |

that inputs to certain layers (e.g., feed-forward) share identical weights, in contrast to inputs to other layers (i.e., self-attention) which do not share weights. Consequently, this enables a streamlined batch processing flow, wherein layers with shared weights are batch-processed and the batch is momentarily unpacked to process each input serially through the attention layers. ORCA shows that this has a negligible impact on the latency since the inputs to the self-attention layers do not share weights and do not benefit from batching. As such, we follow ORCA and use its selective batching technique.

Also borrowing from ORCA the iteration-level scheduling technique, $S^3$ does not wait until all sequences in the current batch finish generation. Instead, it checks if any sequence in the batch has finished generation at every iteration. This grants $S^3$ higher scheduling flexibility and removes any redundant waiting time. Finally, if a model cannot fit in one GPU, $S^3$ uses pipeline parallelism and shard the models in Transformer layer granularity.

**Supervisor**    The supervisor is in charge of supervising the GPU utilization and handling mispredictions. The supervisor runs in the background to check for the available space in the HBM and passes the information to the scheduler. The scheduler then appends a sequence to the running batch if the available memory is large enough for the sequence.

The supervisor is also responsible for handling mispredictions. In the case of short predictions, the supervisor preempts those sequences that exceed their reserved memory. It monitors the length of the current output sequences and evicts them if they are not finished but used up its reserved memory. It asynchronously moves the current state of those sequences including the KV cache and the generated tokens to the request pool and frees up the GPU memory. Now the K and V matrices are fragmented with blank rows where the evicted KV cache was originally stored in. The supervisor shifts the rows below the blank one so that all rows are stored contiguously. This memory format is required by current libaries [24,25] and also resolves the fragmentation issue. Finally, the supervisor doubles the assigned memory for the evicted sequences to fix the short misprediction.

Finally, the supervisor constantly trains the predictor in the background. It uses the sequences that the predictor mistook to train the predictor so that it can learn from its mistakes. This training time is relatively short and our measurement shows that each training iteration takes 11ms on average while sequence generation takes several seconds or more. This implies that the retraining overhead is less than 10% even if we train the predictor for 10 epochs.

**Putting it all together**    We summarize this section with an explanation of how $S^3$ uses each component to serve a sequence-generation request. First, text-generation requests arrive at the request pool. The predictor predicts the output sequence length of the sequences in the pool. The supervisor runs in the background and checks the current HBM usage. Next, the scheduler uses both the predictions and the available HBM to batch requests for maximum GPU utilization. It finishes its job by scheduling that batch to the GPU which generates the scheduled sequences.

# 4    Evaluation

We show that $S^3$ achieves higher throughput by predicting the output sequence length. It does so by using larger batch sizes hence smaller numbers of iterations. One iteration refers to processing a Transformer model once to generate a token. We also show that $S^3$ can reduce the cost of serving models by using fewer GPUs.

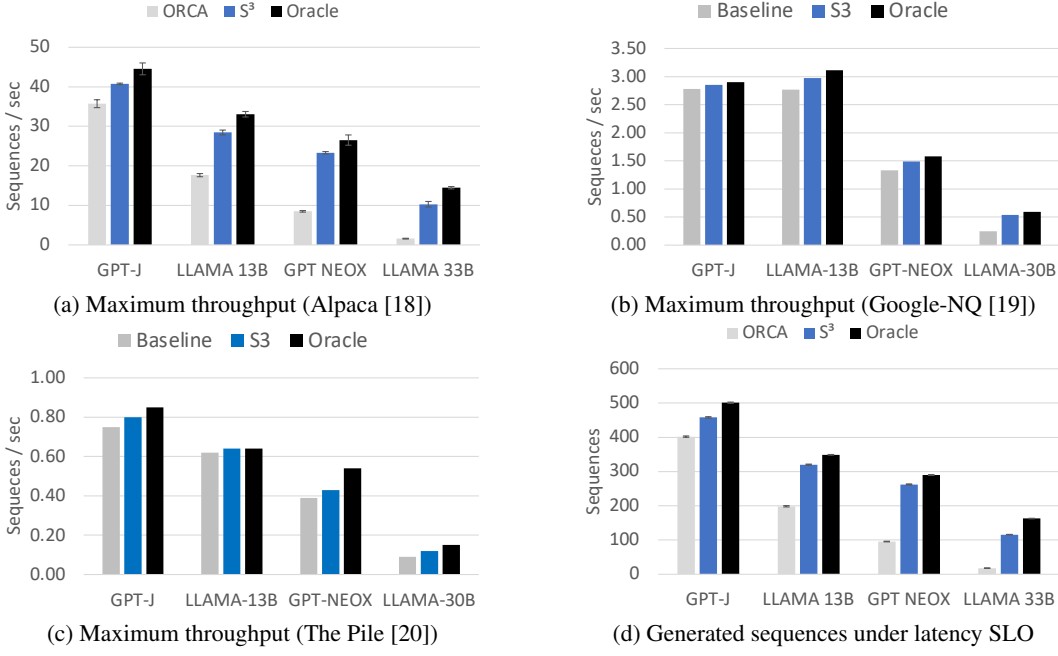

Figure 4: Latency and throughput of different models and datasets.

Table 2: Maximum throughput of the three systems measured in tokens/s

|  | **Baseline** | **S³** | **Oracle** |
|---|---|---|---|
| GPT-J | 2061.16 | 2349.67 | 2569.09 |
| LLAMA-13B | 1018.15 | 1641.55 | 1907.09 |
| GPT-NEOX | 490.15 | 1344.94 | 1530.05 |
| LLAMA-30B | 91.46 | 593.73 | 834.30 |

**Environment**   We run our evaluation on an NVIDIA 80GB A100 GPU connected to the host DRAM via PCIe 4.0×8 in a Lenovo ThinkSystem SD650-N V2 Server [27].

**Baselines**   We compare with ORCA [16], a Transformer serving system that increases throughput by iteration level scheduling and selective batching. ORCA has to allocate the maximum sequence length of memory when it is not aware of the output sequence length to guarantee full sequence generation. S³ predicts the output sequence length and allocates memory based on the prediction. We implement the systems on top of FasterTransformer [4] since this library is faster than HF-Transformers [3] due to more optimizations. We also compare S³ with an ideal system with a perfect predictor which we term Oracle.

**Models**   We use models ranging from 6 billion parameters to 175 billion parameters for the evaluation. The specifics of these models are explained in table 1.

## 4.1   Throughput Analysis

We evaluate S³'s throughput using Alpaca [18], Google Natural Questions (Google-NQ) [19], and The Pile [20] datasets. Specifically, we query S³ with questions and ask it to generate the answers.

**Offline scenario: Maximum throughput**   Fig. 4 (a) - (c) reports the maximum throughput in sequences per second for different models and different datasets. We measure the throughput using the maximum batch size of each configuration. It shows that S³ outperforms ORCA by 1.13× and up to 6.49× and closely matches Oracle, differing from 9.34% and up to 40.52%. The difference is magnified with larger models because the batch size is limited even for S³ since most of the HBM

Table 3: Average batch size and number of iterations for different models

| Model | ORCA | | S³ | | Oracle | |
|---|---|---|---|---|---|---|
| | Batch size | Num iter | Batch size | Num iter | Batch size | Num iter |
| GPT-J | 69.94 | 7988 | 530 | 1054 | 564.88 | 989 |
| LLAMA 13B | 31.61 | 17675 | 274.66 | 2034 | 527.04 | 1060 |
| GPT-NEOX | 16.89 | 33069 | 157.59 | 3545 | 292.19 | 1912 |
| LLAMA 33B | 4 | 139790 | 42.47 | 13155 | 77.35 | 7223 |

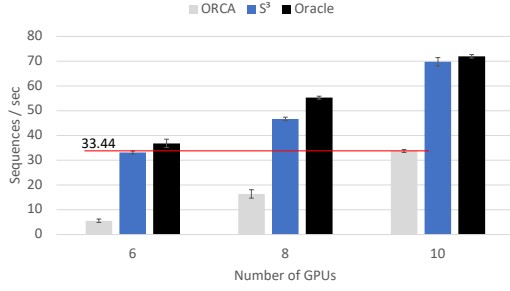

Figure 5: Maximum throughput of GPT3 running on different numbers of GPUs.

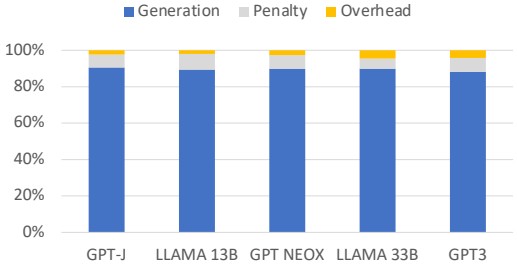

Figure 6: Latency breakdown of S³.

is used to hold model weights. The batch size in S³ gets cut off before saturating the throughput as shown in Fig. 1. We also report the throughput in tokens/s in table 2 evaluated on the Alpaca dataset.

We can notice that the maximum throughput increases by 6.49× while the batch size increases by nearly 10× for every model. This comes from the unique behavior of Transformers. Feed-forward layers in the models benefit from batching layers while self-attention layers do not. This is because inputs in feed-forwards share the same weight while inputs in self-attentions attend to their own sequences. S³'s performance benefit is increased when the parallelized portion is larger than the serialized portion. However, increasing the batch size entails adding more serialized inputs thereby growing the serialized portion. GPT-J is a good example of this characteristic which shows a similar throughput jump of 1.13× from ORCA to S³ and 1.09× from S³ and Oracle while the batch size differs by 8.69× and 1.97×, respectively.

**Online scenario: SLO-aware throughput**  We now consider a scenario with a latency SLO constraint. We set the SLO as 0.1875 seconds to generate one token given that average English readers can read at 4 words per second [12] or 0.25 second per word, and 0.75 words per token [13]. We next calculate the latency SLO for each sequence by multiplying 0.1875 by its sequence length (e.g., 11.25s for a sequence with 60 tokens).

Fig. 4d reports the number of sequences that each model generates using ORCA, S³, and Oracle. Oracle exceeds the SLO for GPT-J, LLAMA 13B, and GPT-NEOX when it chooses the maximum batch size. So we limit the batch size for these models during this evaluation. S³ generates a similar number of sequences with the ideal case and 1.13× to 6.49× more sequences compared to ORCA. The throughput increase is similar to the offline scenario since S³ meets the SLO in most cases with its maximum batch size. However, the difference between S³ and Oracle reduces by 10% compared to the offline scenarios because we limit the batch size hence the throughput of Oracle.

## 4.2 Cost Analysis

We evaluate S³ with different numbers of GPUs. We partition GPT-3 into 6 and 8 GPUs in a pipeline-parallel manner, allocating 16 and 12 transformer layers per GPU for each configuration, respectively. We also evaluate on 10 GPU setting where we allocate 10 layers to 9 GPUs and 6 to the remaining GPU. S³ pipelines each batch and schedules the a batch whenever the GPU processing the first partition passes its result to the next batch. This reduces GPU idle time by having every GPU processing batches concurrently.

Table 4: Average batch size and number of iterations for different system configurations on Alpaca

| Num GPUs | ORCA | | $S^3$ | | Oracle | |
|---|---|---|---|---|---|---|
| | Batch size | Num iter | Batch size | Num iter | Batch size | Num iter |
| 6 | 11.95 | 46734 | 114.22 | 4891 | 209.47 | 2667 |
| 8 | 28.68 | 19482 | 247.85 | 2254 | 470.65 | 1187 |
| 10 | 48.99 | 11403 | 399.62 | 1398 | 564.88 | 989 |

Table 5: Accuracies and runtime of different predictors on different datasets

| Model accuracy (%) | Model size | Datasets | | | Runtime (ms) |
|---|---|---|---|---|---|
| | | Alpaca [18] | Google [19] | The Pile [20] | |
| MS-minibert | 22M | 98.06 | 77.99 | 60.1 | 2.3 |
| Distilbert-base | 66M | 98.6 | 82.68 | 65.6 | 4.1 |
| Bert-base | 110M | 99.54 | 85.08 | 68.2 | 7.6 |
| Bert-large | 340M | 99.6 | 89.25 | 71.9 | 14.5 |

Fig. 5 reports the maximum throughput using the different numbers of GPUs. First, we can see that $S^3$ achieves similar throughput using 6 GPUs compared to ORCA with 10 GPUs. ORCA shows 0.92% higher throughput than $S^3$ to be specific. More GPUs shard the model into finer pieces and leave more space for storing the KV cache, allowing us to increase the batch size. Table 4 supports this claim by reporting larger batch sizes with more GPUs. $S^3$ can achieve similar effect with fewer GPUs by optimizing memory allocation strategy.

Naturally, it leads to a similar question with the one in the throughput evaluation on why $S^3$ with 6 GPUs shows similar throughput with ORCA with 10 GPUs even with $2.33\times$ the batch size. The answer is in the pipelined execution and the sequential nature of self-attention layers in Transformer-based models as explained in 4.1. The systems complete a batch at every $\lceil \frac{l}{number\ of\ GPUs} \rceil$ instead of at every $l$ layers. For example, $S^3$ with 6 GPUs completes a batch at every 16 layers instead of 96 for GPT3. Similarly, ORCA with 10 GPUs completes at every 10 layers and thus processes more quickly. The increase in the latency negates the throughput benefit from $S^3$ such that the two systems show almost identical throughput even with the $2.33\times$ larger batch size when using $S^3$.

### 4.3 Overhead: Latency Breakdown

We evaluate runtime latency of each component in $S^3$. We classify the runtime latency into three categories: generation, penalty, and overhead. Generation is the time $S^3$ spent on processing the model to generate tokens. Penalty denotes the time it took for $S^3$ to preempt the KV cache and hidden states from the GPU and load it back to the GPU. Overhead includes the time it took for predictor, scheduler, and supervisor, combined. Fig. 6 show that penalty and overhead combined are negligible (i.e., 11% on average) compared to the generation. Of course, the penalty would increase if the predictor is less accurate and triggers more data traffic between the GPU and the host. In contrast, the overhead will increase if we employ a more accurate but heavier predictor, thus introducing a new trade-off.

### 4.4 Predictor Ablation Study

We vary the predictor size from 22M to 340M parameters and report their accuracies on different datasets in table 5. We can observe a similar trend in all three datasets where a larger predictor generates more accurate predictions. The accuracies differ among different datasets since the length distribution differs. Specifically, Alpaca [18] showed the smallest variance among the length distributions in the three datasets compared to The Pile [20] showing the greatest.

## 5 Related Works

**Machine learning serving systems** The high interest in machine learning has sparked numerous research in its service platforms [1, 4, 6, 16, 28–38]. Especially, the surprising performance of Transformer-based language models has directed many researchers to develop Transformer-specific

serving systems [4, 6, 34, 37–39]. Most of the systems focus on reducing the latency without much concern for throughput with the exceptions of [6,31]. The throughput-oriented systems use memory hierarchy to store parts of the model in slower memory and to increase the batch size. However, they all allocate the same memory to every sequence and do not consider preempting a sequence based on a prediction. $S^3$ can improve these systems by reducing the required memory size, removing the need for larger but slower memories such as SSDs, and reducing the fiscal cost of memory overall.

FastServe [39] tackles the head-of-line blocking problem caused by query-level scheduling while $S^3$ addresses the inefficient memory allocation issue. It proactively manages KV cache similar to $S^3$'s supervisor by migrating sequences between the host memory and the GPU HBM. However, FastServe does not have an output sequence length predictor and thus uses a skip-join Multi-Level Feedback Queue since it is unaware of the job execution time. We expect $S^3$ to work with FastServe so that $S^3$'s predictor delivers more information to FastServe for better scheduling.

**Reducing the KV cache overhead**  The issue of attention layers in Transformers requiring quadratic computation and memory with respect to the sequence length has been extensively studied. Various approaches have been proposed to address this problem. Low-rank approximation [40, 41] and exploiting sparsity [42–44] are among the methods that aim to mitigate the issue. Another approach, known as multi-query attention [37, 45], suggests reducing the cache size by utilizing one attention head per key-value (KV) cache instead of employing multiple attention heads, as illustrated in Table 1. Additionally, works focusing on model size reduction through compression [46] and quantization [6, 47, 48] can also contribute to reducing the size of the KV cache. These approaches are complementary to our work and can be employed to reduce the penalty caused by mispredictions, allowing for the use of a less accurate but lighter predictor.

**Sequence length prediction**  While there are limited works that predict output sequence length based on an input sequence, Yan et al. [49] propose a convolution-based small network with embedding layers to forecast output sequence length in machine translation. In our approach, we employ a similar strategy but utilize a small Transformer-based predictor to estimate the output sequence lengths in text generation tasks. This predictor allows us to accurately predict the output sequence length for text generation workloads.

## 6 Limitations and Conclusion

**Limitations**  We make the following assumptions in this work. We assume that text generation request traces mimic the publicly available question-and-answering datasets since there are no publicly-available traces. Analyzing the actual trace would facilitate deploying $S^3$ in commercial workloads. Similarly, text generation task does not have any standardized latency SLO constraint as in other machine learning workloads [50] since it is a relatively new service. So we assume average reading speed of an English reader as our SLO. We will be able to evaluate $S^3$ in broader scenarios if organizations publicly release different SLOs for different text generation applications.

**Conclusion**  In summary, we introduce $S^3$, a framework designed to achieve high throughput in serving Transformer-based generative models. $S^3$ leverages a predictor to estimate the output length of generated sequences and schedules them accordingly to maximize throughput. Additionally, $S^3$ handles potential prediction errors to guarantee reliability. By allocating varying memory sizes to different inputs, $S^3$ acknowledges that not all sequences should be treated equally. This approach expands the conventional trade-off between latency and throughput frontier, paving the way for new possibilities in optimizing the latency-throughput trade-off.

## 7 Acknowledgement

We thank the anonymous reviewers for their thoughtful comments and suggestions. This material is based upon work supported by the National Science Foundation under Grant No. 1704834 and No. 2118985 and supported in part by the Application Driving Architectures (ADA) Research Center and the National Science and Technology Council, Taiwan, under Grant No. 112-2222-E-A49-002-MY2. Chun-Feng Wu acknowledges the support from the Yushan Young Scholar Program by the Ministry of Education (MOE) in Taiwan.

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
