# OpenReview forum: "$S^3$: Increasing GPU Utilization during Generative Inference for Higher Throughput"
_NeurIPS.cc/2023/Conference — NeurIPS 2023 poster_

### Official Review · Reviewer_RWv5 · 2023-06-24

**Soundness:** 4 excellent
**Presentation:** 3 good
**Contribution:** 2 fair
**Rating:** 5
**Confidence:** 4

**Summary:**

This work builds an LLM inference platform, called S^3, around a sequence length predictor.
The sequence length predictor is used to
1. batch LLM generations
2. pre-allocate kv cache (where all seq have similar predicted seq len)
S^3 also has a method for handling seq len prediction errors. They do pipelined generation. They unpad shorter seq to ignore wasting compute.

**Strengths:**

LLM inference is becoming a larger and larger part of the total compute used within large organizations. Decrease inference costs is an extremely impactful line of work.

The analysis of overheads is good.

**Weaknesses:**


For kv cache pre-allocation, the authors note that pre-allocation limits seq len, not pre-allocating kv cache slows generation. An obvious baseline is pre-allocating for S tokens at a time (eg S=64) and freeing kv cache mem of generations which have finished (this frees mem for the really long generations). This simple baseline would decrease the kv cache concat overhead by a factor of S and eliminates the overhead of the S^3 algo.

The work uses a sequence length predictor to build an LLM inference platform. This is great systems research, I'm not sure it's positioned well for NeurIPS.
The work doesn't show how much of the benefit comes from pipelining vs kv cache preallocation vs seq len batching vs unpading. I'm not sure how much improvement each method is or if we're introducing other overheads by including methods such as seq len prediction.

There are great works on improving LLM inference. Why only compare to ORCA?



**Questions:**

The authors note: "More GPUs shard the model into finer pieces and leave more space for storing the KV cache, allowing us to increase the batch size. " - line 280
The method also pipelines generations; kv caches for all generations must be stored in mem, this in turn lowers the possible batch size. Is it possible to quantify the trade off?

Will the code be made public? People can easily benchmark the implementation helping the authors identify shortcomings and improve the overall performance over time.


**Limitations:**

Requires training a seq len predictor which hopefully generalizes to different data distributions.

---

> ### Author Rebuttal · Authors · 2023-08-10
>
> - **How does $S^3$ perform against pre-allocating S tokens at a time?**
>
> Table below shows the throughput compared to the reviewer’s recommended system with S=64. Compared to $S^3$, pre-allocation of S tokens allows larger batch size but at the cost of more frequent evictions. That is the scheduler batches assuming that every sequence has input_sequence_length + S tokens at the start. Thus it is able to batch more sequences when S is smaller than the actual generated sequence. However, in such a case, the GPU HBM becomes full more frequently and more sequences are evicted to the host memory and loaded back to the HBM, generating more data traffic.
>
> ||GPT-J|LLAMA-13B|GPT-NEOX|LLAMA-30B
> |-|-|-|-|-|
> Pre-allocate|26.57| 15.32 | 12.1 | 8.39
> $S^3$|40.73|28.46|23.32|10.29
>
> - **Positioning for NeurIPS?**
>
> The emergence of expansive models characterized by extended processing periods and substantial energy consumption has triggered a surge of interest among NeurIPS researchers in the optimization of machine learning (ML) systems. In substantiation of this trend, NeurIPS convened a series of ML systems workshops in 2022 [55, 56], reflecting the growing emphasis on system-level enhancements within the field. Furthermore, a slew of ML systems papers have graced the discourse in recent times. Notable examples include Nimble [57], MCUNet [58], and FlashAttention [59], which garnered attention by securing slots in the main conference track. The examples, among others, exemplifies the burgeoning focus on crafting and refining ML systems to align with the evolving landscape of large-scale models.
> - **Benefit from each technique in $S^3$ and latency overhead**
>
> The main contribution of $S^3$ is the predictor that predicts the output sequence length and pre-allocating the KV cache. Other techniques such as pipelining, batching, and unpadding are proposed in other works. Specifically, pipeline-parallel execution is used by many works [4, 16, 31, 39] while sequence length batching and unpadding are introduced in ORCA [16]. We compare $S^3$ to ORCA to pinpoint the benefit of our main contribution.
> As the reviewer might already know, we want to re-iterate that the overhead introduced by $S^3$’s predictor, scheduler, and supervisor are 11% on average, combined.
> - **Why only compare to ORCA?**
>
> We compared $S^3$ to ORCA since it was the state-of-the-art deployment system at the time of writing. However, ORCA is not opens-sourced so we implemented ORCA on top of FasterTransformer [4], an accelerated language model processing library by NVIDIA. Thus we are comparing $S^3$ to both ORCA and FasterTransformer by comparing to our implementation of ORCA.
> - **Tradeoff of pipelining and memory consumption**
>
> The reviewer is correct about a large KV cache is one of the main culprits of using smaller batch size. However, please note that not all KV caches for the entire generations should be stored in GPU memory. Rather, the scheduler in $S^3$ ensures that the currently running batch size does not exceed the GPU memory capacity and stores KV cache for those sequences that are not currently running in the host memory as described in Section 3 subsection Length-aware sequence scheduler.
> - **Opensourcing the code**
>
> The code of $S^3$ will be open-sourced if accepted. We look forward to the research community making full use of our work and developing $S^3$ over time.
>
> **New references**
>
> 52. Challenges in Deploying and Monitoring Machine Learning Systems (https://nips.cc/virtual/2022/workshop/49982)
> 53. Machine Learning for Systems (https://nips.cc/virtual/2022/workshop/49961)
> 54. Nimble: Lightweight and Parallel GPU Task Scheduling for Deep Learning
> 55. MCUNet: Tiny Deep Learning on IoT Devices
> 56. FlashAttention: Fast and Memory-Efficient Exact Attention with IO-Awareness

---

> > ### Comment · Reviewer_RWv5 · 2023-08-10
> > **Rebuttal comments**
> >
> > Thank you for running the baseline; I stand corrected, having the predictor and sorting does seem to help substantially.
> >
> > It'd also improve the paper if the authors could show performance on MultiQuery and GroupedQuery (with 8 groups cuz TensorParallelism) attention given these are gaining a lot of popularity (ie they are starting to be used in most new popular networks)
> > If performance on MultiQuery and GroupedQuery Attention is shown, I'd change my rating to a `6: Weak Accept` (assuming I can figure out the UI)

---

> > > ### Author Response · Authors · 2023-08-10
> > >
> > > Thank you for the suggestions to improve the paper. Upon preliminary review of the code, we expect to be able to provide results before the deadline.

---

> > > ### Author Response · Authors · 2023-08-18
> > >
> > > We sincerely appreciate the reviewer's insights aimed at enhancing the quality of our paper. Additionally, we appreciate the reviewer's inclusion of references to the memory optimization strategies gaining popularity: multi-query attention (MQA) and grouped-query attention (GQA). These two memory optimization approaches, which focus on minimizing the footprint of the key-value (KV) cache itself, complement the goal  $S^3$, which addresses the reduction of memory inefficiencies.
> > >
> > > As a result, we have incorporated the throughput evaluation of the newly proposed attention strategies, compared to the throughput of the same strategies enhanced by $S^3$. This comparison is presented in the table below, offering a comprehensive understanding of their relative merits. We used 8 groups for GQA in this evaluation as the reviewer suggested.
> > >
> > > Models |MQA|	MQA + $S^3$|	GQA|	GQA + $S^3$
> > > -|-|-|-|-
> > > GPT-J|	41.75|	42.19|	40.08|	41.17
> > > LLAMA-13B|	33.03|	34.22|	29.81|	33.48
> > > GPT-NEOX|	24.85|	27.21|	20.32|	23.69
> > > LLAMA-30B|	13.86|	15.41|	5.9|	14.12
> > >
> > > The table illustrates that the performance gap between the standalone attention strategies and the same strategies enhanced by $S^3$ increases when using larger models. Larger model sizes and higher KV cache dimensions place higher memory pressure, and efficient utilization of memory becomes more important.
> > >
> > > $S^3$ effectively mitigates memory wastage and alleviates memory-related constraints. Consequently, GQA + $S^3$ can achieve a throughput on par with, or better than that of standalone MQA, as $S^3$ counteracts memory pressure and facilitates GQA's performance, even when dealing with substantial model sizes.

---

> > > ### Author Response · Authors · 2023-08-21
> > >
> > > This is a gentle reminder to the reviewer that we have evaluated $S^3$'s performance on MultiQuery and GroupedQuery Attention as the reviewer requested. Please let us know if you have follow-up questions.

---

### Official Review · Reviewer_s4gr · 2023-07-03

**Soundness:** 3 good
**Presentation:** 3 good
**Contribution:** 2 fair
**Rating:** 6
**Confidence:** 3

**Summary:**

The paper proposes a scheme that increases the throughput during inference on Transformer large language models (LLM). Typically, LLMs require large amounts of memory, for model parameters and for the KV (key/value) cache. The KV cache size depends of the output sequence length, which is not known when inference starts. Some implementations allocate memory for the KV cache in small increments, causing large latency, while others preallocate for up to the maximum output sequence length, causing potential memory waste. The authors propose a method, called S3 (scheduling sequences with speculation), that predicts the output length, and allocates the KV cache accordingly. The predictor is a fine-tuned Distilbert model (66M parameters) with small size and fast prediction time. A scheduler batches requests according to a greedy strategy. A supervisor is in charge of checking GPU utilization and handling mispredictions, while at the same time training the predictor in the background. An experimental analysis shows the flexibility of S3 in offering a trade-off between latency and throughput.

**Strengths:**

Transformers are memory and compute intensive. Designing schemes that make better use of memory, thus increasing the latency and/or throughput is an important topic. The proposed solution is simple, and the predictor ran as part of S3 adds negligible overhead. The experimental results show improvements, generating by up to 6.49 times more sequences compared to other existing system.

**Weaknesses:**

The system will need more analysis in the future, based on real traces (which are not available for research community now), and considering realistic service level objectives.

**Questions:**

1. Could you add to the experimental section results for the case when no KV caching is used?

---

> ### Author Rebuttal · Authors · 2023-08-10
>
> - **Real traces and SLO**
>
> We agree with the reviewer and believe that the paper could be strengthened by using real-world request traces and realistic latency SLOs. However, as mentioned both by the reviewer and in Section 6 Limitations, we acknowledge that the traces are not available to the research community at the time of writing. Additionally, we set the latency SLO in our paper is set as the reading speed of an average human [12]. But this will be strengthened if the research community can agree upon a standard benchmark as in MLPerf [50].
> - **How does $S^3$ perform when no KV caching is used?**
>
> Number of FLOPs for recomputating the key (K) and value (V) matrices increases linearly with respect to the previous sequence length. Additionally, the K and V matrices should be stored in the GPU HBM to process self-attention layers as explained in Section 2. So the regenerated K and V matrices will use the identical amount of memory capacity as KV caches and the memory capacity problem will persist. We will add the throughput evaluation of this configuration to Section 4 in the revised version.

---

> > ### Comment · Reviewer_s4gr · 2023-08-12
> >
> > Thank you for answering my questions, I look forward to seeing the revised version.

---

### Official Review · Reviewer_1WaJ · 2023-07-05

**Soundness:** 3 good
**Presentation:** 4 excellent
**Contribution:** 3 good
**Rating:** 7
**Confidence:** 4

**Summary:**

The paper tackles the problem of predicting the number of generated tokens for transformers in text generation tasks. This will help with better memory allocation and batch size management. The previous systems either used dynamic memory allocation (Hugging Face) which incurs inference overhead, or preallocated memory for over-estimated output length, limiting the batch size. This paper's prediction of output length achieves a larger batch size, thus increasing the throughput. Experiments show a 6.49x improvement in throughput. The predictor is get by training a Distilbert model.

**Strengths:**

1. The idea is simple but effective.
2. The paper is clear, well-written, and easy to read.
3. The evaluation is clear and convincing. It contains various settings, including different models, different hardware setups, online/offline with clear information presented (latency breakdown, batch size).

**Weaknesses:**

1. It has not been discussed that different models can generate outputs with different lengths.
2. The evaluation does not contain different request patterns.
3. Missing the evaluation of throughput in term of token/s.

**Questions:**

1. Figure 1: is it get from generating with 60 token context length? This is not a long context.
2. line 71: "6.49× more sequences" is vague, because sequences can have different lengths.
3. line 212: "The supervisor shifts the rows below the blank one so that all rows are stored contiguously.". How much time overhead does this step cause?
4. Why are you measuring the throughput as sequence/s, not token/s? Sequences have different lengths, which makes sequence/s hard to interpret. I can understand that you want to count how many requests can be served, but maybe also add plots for token/s.
5. When you are using the Alpaca dataset, how did you setup the request distribution (coming time, etc)?
6. How does your method compare to this one "Fast Distributed Inference Serving for Large Language Models" that was released recently?

minor:
line 47: typo before "since most of the HBM is used to..."
line 307: [31, 39] are not focus on throughput.

**Limitations:**

To accurately predict the output length, the distribution of the test trace should be similar to the data used to train the predictor. Also, different models will generate answers with different lengths, which requires the training data to reflect a similar behavior to the model used in the generation.

---

> ### Author Rebuttal · Authors · 2023-08-10
>
> - **Do different models generate sequences with different lengths?**
>
> Different models showed similar output sequence length distributions that were skewed toward short sequences.
> - **The rationale behind using sequences/s.**
>
> We wanted to provide the intuition of throughput in terms of how many more requests can $S^3$ process compared to other systems. This metric is similar to job completion time used in “Fast Distributed Inference Serving for Large Language Models” [51]. However, we will add throughput evaluation in terms of tokens/s on the additional page. Please find the result in tokens/s below.
>
>
> |             | Baseline | $S^3$ | Oracle |
> |---------|------------|----|---------|
> GPT-J| 2061.16  |2349.67|2569.09|
> LLAMA-13B|1018.15|1641.55|1907.09|
> GPT-NEOX|490.15|1344.94|1530.05|
> LLAMA-30B|91.46|593.73|834.30|
>
>
> - **Are the results in Fig.1 from generating 60 tokens?**
>
> Yes, the results in Fig.1 are from generating 60 tokens. We used this sequence length as it is the sequence length used by Google in their paper [37] when measuring the latency and throughput of different transformer-based language models.
> What is the request distribution?
> Due to the unavailability of real-world request traces, we assumed that every request is queued in the request pool. The objective of the experiments was to evaluate the performance of $S^3$ during high pressure and to observe the maximum batch size that $S^3$ can generate during such a scenario.
> - **How does $S^3$ compare to “Fast Distributed Inference Serving for Large Language Models”?**
>
> FastServe, the system proposed in the paper “Fast Distributed Inference Serving for Large Language Models” and $S^3$ share similar approaches but with different motivations. $S^3$ and FastServe both use iteration-level scheduling and host-memory offloading. While $S^3$ addresses the inefficient memory allocation issue, FastServe tackles the head-of-line blocking problem caused by query-level scheduling in current LLM serving systems with focus on the scheduler.
> FastServe introduces two contributions, a skip-join Multi-Level Feedback Queue (MLFQ) and proactive KV cache management systems. The proactive KV cache management system is similar with $S^3$’s supervisor in that both of them preempts sequences that have low priority to the host memory and loads them back on the GPU HBM when there are enough available memory. The skip-join MLFQ is used since FastServe does not have an output sequence length predictor and thus does not know the job execution time. It only uses the input sequence length as a guide to set the priority in its skip-join MLFQ. We expect $S^3$ can work with FastServe so that the predictor can give more information to FastServe’s skip-join MLFQ to further accelerate language model serving systems.
> We will add this in the related works section in the camera-ready version if accepted.
> - **Minor typos**
>
> We will fix the typos in the revised version.
>
> **New references**
>
> 51. Fast Distributed Inference Serving for Large Language Models

---

> > ### Comment · Reviewer_1WaJ · 2023-08-16
> >
> > Thanks for the authors' response! I preserve the original score. The paper is overall clean and good.
> > For the question "Do different models generate sequences with different lengths?", I was saying that there are models like GPT4 tend to output wordy answers, while others like alpaca like to answer short. So that a predictor needs to be trained for each model?

---

> > > ### Author Response · Authors · 2023-08-16
> > >
> > > Thank you for your interest in $S^3$ and for clarifying your question. Indeed, your observation is accurate, as our approach involves training distinct predictors for each individual model within a given serving system. For instance, OpenAI's ChatGPT should train separate predictors for both GPT-3.5 and GPT-4. Analogously, serving systems utilizing Alpaca or Vicuna as their generative models should train specific predictors corresponding to each respective generation model. Please note that these models exhibit a continuous learning process through ongoing interactions with users in an online environment.

---

### Official Review · Reviewer_945f · 2023-07-15

**Soundness:** 2 fair
**Presentation:** 3 good
**Contribution:** 3 good
**Rating:** 5
**Confidence:** 3

**Summary:**

This paper presents a new solution to the challenges of GPU underutilization and increasing batch size in the generation task of Large Language Models (LLMs), rooted in the memory-intensive requirement to retain the K and V values of prior tokens. To tackle these issues, the paper proposes an efficient framework named S^3, which uses a fine-tuned Distillbert model as a predictor to forecast the output sequence length based on an input prompt. This predictive model guides query generation scheduling and manages mispredictions by halting sequences exceeding allocated memory and doubling their assigned memory. The proposed approach increases the maximum configurable batch size in both online and offline scenarios and improves throughput, reporting a throughput increase of up to 6.49 times. Although this paper offers improvements in handling memory allocation for LLM inference, there are some concerns about evaluation and ablation of the proposed techniques.

**Strengths:**

- This paper tackles the vital challenge of predicting output sequence length through a unique DistillBERT-based co-optimization of the system and algorithm. This approach effectively addresses the limitations of existing frameworks, namely HF-Transformer and FasterTransformer, leading to improved throughput while meeting SLO latency.

- While larger models typically yield better performance, as the space for K, V caches reduce, effective memory allocation management becomes increasingly essential. The S^3 framework notably elevates the maximum throughput, particularly in larger models.

- The S^3 framework optimizes the latency-throughput trade-off by adjusting memory allocation, leaving the model architecture intact and thereby maintaining model perplexity. Operating independently and focusing on performance enhancement, the framework doesn't impact the accuracy of existing LLM models.



**Weaknesses:**

- While a major contribution of the paper is the prediction of output sequence length, the effectiveness of this proposed predictor isn't thoroughly evaluated. Tables 2 and 3 show a significant difference between the batch size predictions of S^3 and Oracle. The reasons behind this discrepancy need to be explained.

- The effectiveness of the predictor appears to vary greatly with the fine-tuning dataset, with accuracies ranging from 65.6% to 98.6%. The paper proposes online learning, but this isn't evaluated. Additionally, the paper lacks an ablation study on the choice of predictor models and sizes.

- The paper only compares the latency-throughput trade-off between vanilla systems and S^3 and doesn't compare average throughput and latency in specific tasks. This is crucial, as output sequence length prediction accuracy, which seems to be task-dependent, would likely impact this.

- While S^3 meets the SLO limit, its implementation generally results in an increase in end-to-end latency. The paper doesn't adequately address how much each proposed technique contributes to this overhead.

- The paper introduces a supervisor component to handle mispredictions, but the description of this technique is unclear. The supervisor appears to allocate double the memory when a previous allocation isn't sufficient. However, in typical scenarios with scarce memory, this could lead to problems if the predictor misestimates the batch size. More detailed explanations of various scenarios would be useful.


**Questions:**

- The figure captions could use more detail. For instance, is Figure 1 (left) showing an online scenario, and how many GPUs were used? Similarly, does Figure 1 (right) represent an offline scenario, and how large is the model size?

- The term "number of iterations" requires clarification. It's evaluated in the sections assessing performance, but isn't clearly defined.

- In Section 4.1, it's mentioned that "Oracle exceeds the SLO for GPT-J, LLAMA 13B, and GPT-NEOX when it chooses the maximum batch size". It's unclear why Oracle would select a batch size that violates the SLO.

- Both Figures 4 (a) and (b) claim a maximum performance improvement of 6.49x. How can this be, especially when Figure 4 (b) pertains to "Generated sequences under latency SLO"? Is the data in Figure 4 (b) based on a scenario of maximum throughput?

- The application of S^3 seems to yield less throughput improvement as the model size decreases. Is there still a significant performance improvement when applying S^3 to smaller models like GPT-3 small, which has around 125M parameters?

- Regarding throughput-related performance metrics, it's unclear how Oracle's (ideal predictor and scheduler) performance was measured. The scheduling might have been done under an entirely ideal scenario, or it might be based on the assumption that the predictor makes no errors in predicting input sequences.

**Limitations:**

The authors properly stated the limitations of the proposed methods.

---

> ### Author Rebuttal · Authors · 2023-08-10
>
> - **The discrepancy between the batch sizes in $S^3$ and Oracle**
>
> The predictive mechanism of $S^3$, elaborated upon in Section 3 under the Predictor paragraph, revolves around forecasting the length bucket. $i^{th}$ bucket contains sequences with lengths in [max_seq_len/num_buckets * i, max_seq_len/num_buckets*(i+1)). Consequently, even when the predictor attains accurate answers, a potential variance of up to max_seq_len / num_bucket remains. Nevertheless, our throughput assessments validate $S^3$'s efficacy in capitalizing on a substantial batch size that saturates utilization of available compute resources. Fig. 1 demonstrates the saturation and throughput evaluation in Section 4 reports the small throughput difference.
>
> - **Ablation study on the predictor**
>
> We agree with the reviewer that the ablation study will strengthen the paper. The accuracy results are shown in the table below. Larger models result in higher accuracy but longer latency. We chose Distilbert by eliminating other models. Specifically, MS-minibert had low accuracy while Bert models were too large compared to their accuracy gain.
>
> |Model accuracy|Model size|Alpaca|Google|Pile|
> |-|-|-|-|-|
> MS-minibert|22M|98.06|77.99|60.1
> Distilbert-base|66M|98.6|82.68|65.6
> Bert-base|110M|99.54|85.08|68.2
> Bert-large|340M|99.6|89.25|71.9
>
> - **Evaluation on different tasks**
>
> We report the throughput of $S^3$ on different tasks including text retrieval [19] and web scrape [20]. The benefit of $S^3$ decreases when tested on web scraping. There are two main reasons: 1) lower predictor accuracy and 2) longer output sequence length. 1) Lower predictor accuracy generates larger data traffic between the GPU and the host, increasing the latency. 2) Longer output sequence length results in smaller memory waste when allocating maximum sequence length of memory to each sequence. Therefore, memory savings by $S^3$ are reduced.
>
> |||Alpaca|Google|Pile|
> |-|-|-|-|-|
> GPT-J|ORCA|35.73|2.78|0.75
> ||$S^3$|40.73|2.85|0.80|
> ||Oracle|44.54|2.90|0.85
> LLAMA-13B|ORCA|17.65|2.77|0.62
> ||$S^3$|28.46|2.97|0.64
> ||Oracle|33.06|3.11|0.64
> GPT-NEOX|ORCA|8.50|1.33|0.39
> ||$S^3$|23.32|1.49|0.43
> ||Oracle|26.53|1.58|0.54
> LLAMA-30B|ORCA|1.59|0.25|0.09
> ||$S^3$|10.29|0.54|0.12
> ||Oracle|14.46|0.59|0.15
>
> - **Overhead breakdown of $S^3$**
>
> We merged the latency overhead attributed to each constituent element within the $S^3$ in Fig. 6 due to two reasons. Firstly, the observed overhead across the components was found to be inconsequential; secondly, it's imperative to emphasize that the effectiveness of $S^3$ hinges upon the collective utilization of all its components. To elucidate, using solely the predictor, without the scheduler or the supervisor, or the exclusive deployment of the supervisor in isolation, fails to yield any discernible advantages or enhancements.
> - **More detailed role of the supervisor**
>
> The reviewer is correct that the supervisor allocates double the previously allocated memory when the previous allocation is insufficient. However, we found out that doubling the initial prediction resolved most of the mispredictions in the evaluated task.
> - **Details in Figure 1 caption**
>
> The left figure portrays an online scenario, where a single GPU is allocated for each model. This setup assesses the impact of $S^3$ on language models that remain within the size confines of a single GPU. Conversely, the right figure illustrates an offline scenario, wherein the GPT-3 175 billion parameter model is partitioned across a varying number of GPUs. This configuration aims to show the performance of $S^3$ within a distributed computational environment. We will augment the caption in the revised version for clarity.
> - **Clarify the term “number of iterations”**
>
> One iteration refers to processing a Transformer model once to generate a token as explained in Section 2 L87. We will reiterate this at the start of Section 4.
> - **Why does Oracle choose a batch size that violates the SLO**
>
> We designed the system with Oracle to maximize throughput. We implemented the system so that Oracle knows the output sequence length prior to the actual generation, without additional difference with $S^3$. Therefore, it will maximize its batch size in order to maximize the throughput, hence violating the SLO.
> - **How does throughput enhancement under latency SLO match the one without the SLO constraint?**
>
> Generated sequences under the latency SLO in Fig. 4 (b) were counted when ORCA, $S^3$, and Oracle systems maximized their throughput. Sequences that exceeded the latency SLO were ignored during the count. Both Fig. 4 (a) and Fig. 4 (b) show the same maximum throughput increases. This is because the largest batch size that $S^3$ could utilize for LLAMA 33B model was small enough such that the latency of processing requests fell below the latency SLO. We direct the reviewer to Fig. 1 (a) where the SLO is longer than the execution latency of the maximum batch size when using $S^3$.
> - **How does $S^3$ apply to smaller models?**
>
> The reviewer is correct in that $S^3$’s benefit is diminished when using smaller models. This is because smaller models are able to utilize large enough batch sizes to saturate the GPU computation resources without efficient memory management due to the small model and KV cache sizes. However, using GPUs with smaller HBM will cause the same issue of memory scarcity and $S^3$ will help in a similar manner. As the trend toward ML models is enlarging the model size, we believe that efficient memory utilization will become more important.
> - **How was Oracle’s performance measured?**
>
> We evaluated Oracle’s performance by only switching the predictor in $S^3$ with a predictor that always makes the correct prediction. We first ran a profile process that measured the output sequence lengths. Then we ran the actual evaluation where we measured the latency and throughput assuming that the generation models generate identical output as the models in the first profiling process.

---

> > ### Comment · Reviewer_945f · 2023-08-19
> > **Thank you for the answers**
> >
> > I appreciate the authors' response to my questions, which resolved many of my concerns. But still, there are remaining concerns that the analysis of the effectiveness of the proposed predictor was mostly empirical. Thus, I would like to maintain my original rating.

---

### Official Review · Reviewer_CH64 · 2023-07-25

**Soundness:** 3 good
**Presentation:** 3 good
**Contribution:** 3 good
**Rating:** 6
**Confidence:** 2

**Summary:**

The paper proposes a simple yet effective systematic solution to increase GPU usage and throughput during inference. The authors first make an interesting observation that the existing large language models are bounded by memory and the inefficiency is caused by the lack of awareness of the sequence length. Accordingly, the authors first fine-tune a small model DistillBert to predict the sequence length. Based on it, the authors accordingly schedule the generation of queries and deal with mispredictions. The authors conduct experiments with models from 6 billion to 175 billion parameters to evaluate the effectiveness of the proposed system.

**Strengths:**

++ The paper solves a practical problem, personally, I think improving the GPU utilization can benefit the community and the deployment of different applications.

++ The proposed solution is simple yet effective. Predicting the output sequence length first is a simple and intuitive way to schedule the network and better utilize the GPU. Although incorporating a new model introduces additional lags, the authors show that such overhead is negligible in Figure 6.

++ The results look promising; the proposed method allows for larger inference batch size and during the inference, e.g., 32 -> 256 for LLaMA-13B and 4 -> 32 for LLaMA-32B, and hence achieves higher throughput.


**Weaknesses:**

-- Subsection Length-aware sequence scheduler in Section 3 is not very intuitive to the readers, especially the ones that are not directly working on the same topic. It would be better to illustrate more on how the bin packing problem is solved and the batching technique in ORCA (with an Algorithm or diagram or more descriptions).

-- The authors only evaluate the proposed method on NVIDIA A100 GPU. However, it seems the design is not dependent on the architecture of A100 GPU. It would be better to also evaluate the proposed method on other GPU architectures.

-- Minor: Distilbert in L169 and Distil-bert in L155

**Questions:**

See weakness.

---

> ### Author Rebuttal · Authors · 2023-08-10
>
> - **More details of $S^3$ in Section 3 Subsection Length-aware sequence scheduler**
>
> In addressing the bin-packing problem, we leveraged the decreasing first-fit algorithm, which orchestrates bin-packing by prioritizing the largest to smallest load sequence. Notably, $S^3$ orchestrates its packing strategy by queuing the lengthiest sequences first, reserving room for shorter sequences within the GPU's available HBM. This approach has been consciously adopted due to its minimal associated overhead, while still maintaining an approximately-optimal resolution for the problem at hand.
> In a distinct vein, ORCA introduces a novel technique termed selective batching, explained upon in the range of lines 190 to 196. Within the layers of a Transformer model, inputs to feed-forward layers share identical weights. However, attention layers prohibit weight sharing among inputs. In a shrewd maneuver, ORCA discerns this challenge within Transformer-based models and strategically batch processes layers amenable to weight-sharing inputs. Consequently, this enables a streamlined batch-wise processing flow, wherein the batch is momentarily unpacked to process each input serially through the attention layers. This output is subsequently packed into a batch configuration, facilitating collective processing within non-attention layers.
> As we endeavor to refine and enhance the comprehensibility of our submission, we will incorporate the elucidated insights into the camera-ready version, ensuring a more intuitive and accessible presentation for the esteemed readership upon its potential acceptance.
> - **$S^3$’s performance on different GPUs**
>
> We evaluated $S^3$ on NVIDIA A100 GPUs since they were the accessible state-of-the-art GPUs. We could not get access to H100 due to its global hardware shortage. Similarly, we could not evalaute on AMD GPUs due to their limited programmability.
> - **Minor: Distilbert in L169 and Distil-bert in L155**
>
> We will use the term Distilbert throughout the revised version of the manuscript.

---

> > ### Comment · Reviewer_CH64 · 2023-08-16
> > **Thanks for the feedback and a follow-up question**
> >
> > I thank the authors for their feedback!
> >
> > I still have a follow-up question. Is the authors' implementation dependent on GPU types? If so, which feature of NVIDIA A100 GPU is necessary and can your implementation work on other popular GPU types, e.g., V100 or GeForce 3090 GPUs?

---

> > > ### Author Response · Authors · 2023-08-18
> > >
> > > We appreciate your insights that enhance the assessment of $S^3$. Just as the initial reviewer underlined in the original review, $S^3$ stands as a versatile system design that remains independent of specific GPU hardware. Our implementation adheres to this concept, ensuring compatibility with a range of GPUs, including the V100 GPU.
> > >
> > > In response to your query, we've included a table below that showcases the throughput advantages (measured in sequences per sequence) when $S^3$ is employed on a V100 GPU. This data should provide a clear perspective on the performance gains achievable within this context. We sincerely appreciate your engagement and are delighted to provide this additional context for your consideration.
> > >
> > > |V100|ORCA|$S^3$|Oracle|
> > > |-|-|-|-|
> > > |GPT-J|10.45|22.36|25.73|
> > > |LLAMA-13B|1.42|9.4|14.19|
> > >
> > > Please note that we only evaluate using the two smallest models in our evaluation section since larger models do not fit on a 32GB V100 HBM.

---

> > > > ### Comment · Reviewer_CH64 · 2023-08-19
> > > >
> > > > Thanks for the authors' response which solves most of my concerns. Thus, I keep my rating as accept.

---

### Author Rebuttal · Authors · 2023-08-10

We extend our sincere gratitude to the reviewers for their thoughtful and insightful feedback on our NeurIPS submission, titled “$S^3$: Increasing GPU Utilization during Generative Inference for Higher Throughput.” Their meticulous evaluation and constructive comments have undeniably enriched the quality of our work. We highly appreciate the time and effort invested by the reviewers in meticulously assessing our manuscript. In this rebuttal, we address each of the reviewers' comments and suggestions comprehensively, outlining the revisions and enhancements we will make to our paper in response. Through this iterative exchange of ideas, we aspire to contribute to the advancement of knowledge in the field while maintaining the rigor and integrity of our research. Thank you once again for your invaluable contributions to our work.

---

### Decision · Program_Chairs · 2023-09-21

**Decision:**

Accept (poster)

**Comment:**

Overview: The paper proposes a simple yet effective systematic solution to increase GPU usage and throughput during inference. The authors first make an interesting observation that the existing large language models are bounded by memory and the inefficiency is caused by the lack of awareness of the sequence length. Accordingly, the authors first fine-tune a small model DistillBert to predict the sequence length. Based on it, the authors accordingly schedule the generation of queries and deal with mispredictions. The authors conduct experiments with models from 6 billion to 175 billion parameters to evaluate the effectiveness of the proposed system.

Pros:

- Instead of listing, the positive points are mostly within all the reviews included in the reviewing process. Interesting Problem Setting, Simplicity of approach, Thorough literature review, Interesting results.

Cons:

- Only disadvantage the lack of exhaustive results, that was handled during the discussion.

Overall: Based on the initial reviews + discussion with the authors, the paper has only positive comments, and all the comments raised by the reviewers were adequately tackled by the authors.

The only requirement for the authors is to include (if possible and if space allows) any additional discussion that is useful during the rebuttal phase, in order to improve the paper. Please do not include material that is not presented during the rebuttal and material that cannot be checked, unless another round of review is required.